# DNA Hypomethylation Is Associated with the Overexpression of INHBA in Upper Tract Urothelial Carcinoma

**DOI:** 10.3390/ijms23042072

**Published:** 2022-02-13

**Authors:** Chien-Chang Kao, Yin-Lun Chang, Hui-Ying Liu, Sheng-Tang Wu, En Meng, Tai-Lung Cha, Guang-Huan Sun, Dah-Shyong Yu, Hao-Lun Luo

**Affiliations:** 1Division of Urology, Department of Surgery, Tri-Service General Hospital, National Defense Medical Center, Taipei 11490, Taiwan; guman2011@gmail.com (C.-C.K.); doc20283@mail.ndmctsgh.edu.tw (S.-T.W.); en.meng@gmail.com (E.M.); tlcha@ndmctsgh.edu.tw (T.-L.C.); ghsun@ndmctsgh.edu.tw (G.-H.S.); yuds45@gmail.com (D.-S.Y.); 2Graduate Institute of Medical Sciences, National Defense Medical Center, Taipei 11490, Taiwan; 3Department of Urology, Kaohsiung Chang Gung Memorial Hospital and Chang Gung University College of Medicine, Kaohsiung 83301, Taiwan; tailanylyl@cgmh.org.tw (Y.-L.C.); ying_1011@hotmail.com (H.-Y.L.); 4Center for Shockwave Medicine and Tissue Engineering, Kaohsiung Chang Gung Memorial Hospital and Chang Gung University College of Medicine, Kaohsiung 83301, Taiwan

**Keywords:** INHBA, upper tract urothelial carcinoma, hypomethylation

## Abstract

Urothelial carcinoma includes upper urinary tract cancer (UTUC) and bladder cancer. Although nephroureterectomy is the standard treatment for UTUC, the recurrence rate is approximately half and the tumor is associated with poor prognoses. Metastases are the most devastating and lethal clinical situation in urothelial carcinoma. Despite its clinical importance, few potential diagnostic biomarkers are suitable for early UC detection. We compared high-stage/high-grade urothelial carcinoma tissues to adjacent normal urothelial tissues using methyl-CpG binding domain protein capture for genome-wide DNA methylation analysis. Based on our findings, inhibin βA (INHBA) might be associated with carcinogenesis and metastasis. Further, clinical UC specimens had significant INHBA hypomethylation based on pyrosequencing. INHBA was detected by real-time PCR and immunohistochemistry staining, and was found to be highly expressed in clinical tissues and cell lines of urothelial carcinoma. Further, INHBA depletion was found to significantly reduce BFTC-909 cell growth and migration by INHBA-specific small interfering RNA. Interestingly, a positive correlation was found between SMAD binding and extracellular structure organization with INHBA using gene set enrichment analysis and gene ontology analysis. Together, these results are the first evidence of INHBA promoter hypomethylation and INHBA overexpression in UTUC. INHBA may affect urothelial carcinoma migration by reorganizing the extracellular matrix through the SMAD pathway.

## 1. Introduction

Urothelial cancer is the fourth most common cancer in the world and is located in the lower (bladder and urethra) or upper (pyelocaliceal cavities and ureter) urinary tract. Bladder cancer is the most common urothelial cancer, while upper urinary tract urothelial carcinoma (UTUC), which is more malignant, accounts for 5% to 10% of cases [1,2,3]. Advanced UTUC is often associated with poor oncologic outcomes [4]. Radical nephroureterectomy (RNU) with bladder cuff excision is the gold standard surgical treatment for patients with UTUC [5,6]. After RNU, a substantial proportion of patients may experience disease recurrence, develop metastases, and subsequently die from the disease [7,8]. Several possible predictive factors for poor prognosis have been reported, including gender [9], stage [10], grade [11], lymphovascular invasion [12,13], and tumor architecture [8,14]. As there are few diagnostic biomarkers for UTUC, the investigation of potential prognostic factors for evaluating prognosis could help predict this high incidence of disease recurrence and promote effective treatment.

Epigenetics includes heritable modifications of chromatin, independent of genetic alterations such as mutations, transversions, transitions, and deletions in DNA. Epigenetic studies involve DNA methylation, post-translational histone modifications, and transcriptional regulation by non-coding RNAs, such as miRNAs and long ncRNAs [15,16,17]. As one of the most abundant and well-studied epigenetic modifications, DNA methylation consists of the covalent addition of a methyl group at the carbon 5 position of the cytosine ring. DNA methylation is an important epigenetic modification in cancer and alters gene expression without changing the DNA sequence, typically hypermethylated tumor suppressor genes [18,19] and hypomethylating oncogenes [20].

Inhibin βA (INHBA) is a transforming growth factor-beta superfamily protein that is a subunit of activin and inhibin. Because inhibins were discovered and isolated before activins, the β subunits were designated inhibin βA and βB [21]. Activin A, the homodimer of two inhibin βA subunits, is the most extensively characterized activin and stimulates follicle-stimulating hormone biosynthesis and release from the pituitary [22]. According to several studies, the overexpression of INHBA is associated with lung, esophageal, colon, gastric, prostate, and urothelial carcinoma [23,24,25,26,27,28]. INHBA has been identified to have a crucial role in malignant biological behavior, including proliferation, migration, invasion, and metastasis [29,30,31,32]. Previous studies revealed that the overexpression of INHBA may be affected by promoter methylation in cancer [23,24,26]. Based on our previously published DNA methylation microarray data from the National Health Research Institute’s (NHRI) bioinformatics database for UTUC [19], high-stage/high-grade UTUC samples had significant INHBA hypomethylation compared to normal urothelium specimens. The objective of this study was to investigate the role of INHBA in UTUC.

## 2. Results

### 2.1. DNA Hypomethylation Genes in Upper Tract Urothelial Carcinoma

The symptoms of UTUC are nonspecific. Further, UTUC is difficult to detect at an early stage. Previously, we reported that SPARCL1 is a hypermethylated gene and a tumor suppressor-like gene in the NHRI bioinformatics database for UTUC [19]. To study potential cancerous factors in UTUC, we re-explored the same bioinformatics database using a methyl-CpG binding domain (MBD) protein microarray to compare the methylation differences between the high-grade and high-stage urothelial tumor samples and normal urothelium samples. We obtained renal pelvis tumor tissue from patients with solitary renal pelvis urothelial carcinoma and normal urothelial tissue from at least 5 cm away from the tumor site (Figure 1A). The results were evaluated through principal component analysis, which allowed the identification of differences in DNA methylation of the tumor and normal tissues (Figure 1B). We identified 260 hypo-methylated differentially methylated regions (hypo-DMR), defined as regions with a model-based analysis of tiling arrays (MAT) score > 7 and *p* values < 0.05. Based on GO analysis, hypo-DMRs were significantly enriched in biological processes, including the multicellular organismal process, response to stimulus, single-organism process, cellular component organization or biogenesis, growth, development process, and others (Figure 1C). Of the various categories, those involving growth are crucial and indispensable for carcinogenesis and have not been systemically evaluated in UTUC. As shown in Figure 1D, Venn diagram analysis established the following conditions: hypomethylation of the promoters of genes in tumor tissues relative to normal tissues, upregulation of genes in tumor tissues relative to normal tissues, high expression and poor survival, and growth identified from gene ontology analysis. INHBA was the only candidate that met these criteria. INHBA overexpression may be affected by promoter methylation in lung adenocarcinoma and esophageal adenocarcinoma [23,24]. Overexpression of INHBA has been reported in several malignant tumors such as lung cancer, esophageal adenocarcinoma, gastric cancer, prostate cancer, and urothelial carcinoma [23,24,25,26,27,28]. However, its regulatory mechanism affecting tumor cells is not clear. Therefore, we opted to further explore the mechanism of INHBA in UTUC.

### 2.2. Association between the Methylation and Expression of INHBA and Clinical Characteristics of Bladder Cancer Patients

We analyzed data on bladder cancer (BLCA), which is the same as that of UTUC belonging to urothelial carcinoma, to further understand the importance of INHBA in clinical prognosis via the UALCAN web portal [33] using the TCGA-BLCA database. Promoter methylation was significantly reduced in bladder cancer tissues compared to normal tissues (Figure 2A). The expression level of INHBA was relatively high in tumor tissues (Figure 2B). A comparison between INHBA expression and DNA methylation status suggested that gene expression might be negatively related to some CpG sites. We further explored whether INHBA expression was correlated with tumor stages, nodal metastasis status, histologic grade, histologic subtypes, and molecular subtypes in bladder cancer. Based on tumor stages, INHBA expression was upregulated in the late stage (T3–T4) in Figure 2C. As shown in Figure 2D, increased expression of INHBA was observed in tumor tissues with regional lymph node metastasis (N1–N3) compared to those with no regional lymph node metastasis (N0). INHBA expression was most enhanced in the high-grade group compared to the low-grade group (Figure 2E). INHBA mRNA expression was upregulated in the non-papillary subtypes compared to the papillary subtypes (Figure 2F). Interestingly, basal squamous subtypes displayed higher INHBA expression than the neuronal, luminal, luminal-infiltrated, and luminal papillary subtypes (Figure 2G). Kaplan–Meier analysis revealed the relationship between INHBA expression and survival in patients with bladder cancer. Further, the survival rate was significantly negatively correlated with INHBA expression (Figure 2H). Integrating the above results, our findings indicate that the promoter region of INHBA in bladder cancer is hypomethylated, which may affect its increase in tumor expression, and is associated with poor prognosis.

### 2.3. Hypomethylation and Overexpression of INHBA in Upper Tract Urothelial Carcinoma

The extent of INHBA methylation in patients with UTUC tumors was further validated by pyrosequencing. INHBA had a differentially methylated region located on chromosome 7, with a position of 41707521-41707783, starting from -7534 from the start codon (Figure 3A). The 5′ upstream region of INHBA in the software (PyroMark^®^ assay design SW 2.0) was used to predict two methylation sites in Figure 3B. The methylation status of two CpG sites from 24 paired UTUCs and adjacent urothelium is shown in Figure 3C. As shown in Figure 3D, significant CpG site hypomethylation was found in UTUC tissue compared with the adjacent normal urothelium. Therefore, we sought to further verify whether the expression level of INHBA in tumor tissues was higher than that in non-tumor tissues. Using a real-time PCR to detect INHBA mRNA expression, we found that INHBA was highly expressed in tumor tissues compared to non-tumor tissues (Figure 3E). Therefore, we analyzed INHBA protein expression using immunohistochemistry. Figure 3F shows that the protein expression of INHBA in tumor tissues was higher than that in non-tumor tissues.

### 2.4. INHBA Regulates the Cell Proliferation and Migration In Vitro

We opted to further investigate the role of INHBA in urothelial carcinoma based on the discovery of hypomethylation and the high expression of the INHBA gene in UTUC. Real-time PCR (Figure 4A) analyses of the high-grade urothelial carcinoma cell lines (T24 and BFTC-909) revealed higher INHBA expression levels than low-grade urothelial carcinoma cell lines (RT4 and UM-UC-14). This result is consistent with that shown in Figure 2E. The elevated INHBA expression in UC cell lines led us to investigate whether endogenous INHBA affects the growth capability of UTUC cells. Small interfering RNA was used to knock down INHBA protein expression in the BFTC-909 cell line. Based on the results, INHBA mRNA and protein expression was effectively silenced in BFTC-909 cells at day 3 after transfection (Figure 4B). INHBA depletion was induced in BFTC-909 cells. Thereafter, the alamarBlue assay was performed, which revealed that the suppression of INHBA in vitro decreased cell viability (Figure 4C). Additionally, we investigated whether INHBA plays a role in the metastasis of UTUC. The cell migration was decreased as showed by wound-healing assay in BFTC-909 cells transfected with si-INHBA (Figure 4D). The migration of BFTC-909 cells was suppressed by INHBA knockdown using transwell assay (Figure 4E). Based on the results, reducing the expression of INHBA in BFTC-909 cells inhibited cell proliferation and migration.

### 2.5. Ontology Analysis with INHBA and Co-expressed Genes Reveals Signaling Pathways in Urothelial Carcinoma

To explore the mechanism involved in the regulation of cell proliferation and migration by INHBA (Figure 4), we further explored the differentially expressed genes and signaling pathways related to INHBA in TCGA-BLCA cohort using the LinkedOmics database [34]. The top 50 significant genes positively correlated with INHBA are shown in the heat map (Figure 5A). GO analysis was performed with INHBA and its positively correlated genes using the Enrichr tool [35] to analyze their functions in biological processes and transcription factor protein–protein interactions. The top five GO biological processes for INHBA and its positively correlated genes were mainly related to extracellular structure organization, external encapsulating structure organization, collagen fibril organization, supramolecular fiber organization, and skin development (Figure 5B). INHBA and positively correlated genes were mainly related to SMAD4, STAT5A, ZNF148, SMAD3, and SMAD2 linked with the top transcription factor protein–protein interactions (Figure 5B). Moreover, gene set enrichment analysis (GSEA) of all significantly correlated genes was conducted using the LinkInterpreter module in the LinkedOmics database. As shown in Figure 5C, SMAD binding and extracellular structure organization were positively correlated with INHBA. These data showed that INHBA knock-down in BFTC-909 cells reduced the expression of Smad4, Smad3, and Smad2 (Figure 5D). Therefore, we determined whether INHBA could regulate the relative gene expression of the extracellular structure organization. The suppression of INHBA by siRNA was found to decrease the mRNA transcription of FN1 and COL1A1 in BFTC-909 cells (Figure 5E). Next, we observed that FN1 and COL1A1 expression were significantly correlated with INHBA expression in the TCGA BLCA cohort using the LinkedOmics database (Figure 5F). Such findings indicate that INHBA plays an extensive role in extracellular structure organization.

### 2.6. Analysis of mRNA Expression-Based Molecular Subtypes in Urothelial Carcinoma

Our in vitro study showed that the aggressive behavior of urothelial carcinoma could be reversed by INHBA downregulation based on cell viability and migration assays. Therefore, we opted to further dissect the correlation between the expression of INHBA and the genes that affect aggressive behavior in urothelial carcinoma. The RNA sequencing results for 20 samples were analyzed using data from the NHRI bioinformatics database for UTUC. Furthermore, we analyzed 408 samples from The Cancer Genome Atlas (TCGA) Firehose Legacy cohorts of bladder urothelial carcinoma. Ranking the INHBA gene from high to low expression revealed that its expression was not correlated with gender in UTUC and BLCA. However, INHBA was highly expressed in the late-stage (T3–T4) and had a low expression in the low-grade and papillary histology. The high expression of INHBA might not have a significant effect on cell proliferation-related genes in UTUC and BLCA (Figure 6A). The low expression of INHBA resulted in higher expression of epithelial E-cadherin and lower expression of epithelial–mesenchymal transition (EMT)-related genes, such as vimentin, Twist1, Snail, and N-cadherin (Figure 6B). Furthermore, high INHBA expression was found to have relatively high expression levels of extracellular matrix (ECM) markers (FN1, COL1A1, COL3A1, VCAN, and COL5A1), as shown in Figure 6C. These findings confirm that INHBA repression inhibits cell migration ability (Figure 4) and verify that INHBA is involved in extracellular structure organization, as shown in Figure 5B,C. As shown in Figure 2G, the basal squamous subtype had a higher INHBA expression than the luminal subtypes (luminal, luminal infiltrated, and luminal papillary). We then proceeded to explore the distribution of genes related to basal and luminal markers [36]. In both cohorts, we found that the expression levels of basal marker genes, such as KRT6A, KRT14, and KRT16, were increased in patients with high INHBA expression (Figure 6D). However, most patients with low expression levels of INHBA showed high expression of luminal markers (PPARG, GATA3, FGFR3, FOXA1, and ERBB2), as shown in Figure 6E.

## 3. Discussion

The symptoms of UTUC are nonspecific and UTUC is difficult to detect at an early stage. Although most cases of UTUC are characterized by noninvasive patterns, patients with advanced UTUC have poor oncologic outcomes even when treated with radical nephroureterectomy. As there are few diagnostic biomarkers of UTUC, it is important to identify more potential prognostic factors for evaluating prognosis. Based on increasing evidence in recent decades, epigenetic modifications play an important role in urothelial carcinoma [37,38]. To determine whether there are any epigenetic changes in the tumorigenesis of UTUC, we selected tumors of high-grade and high-stage UTUC and normal urothelial tissue by comparing differentially methylated regions using a methyl-CpG-binding domain protein microarray. The genes involved in growth are crucial for carcinogenesis and have not been well studied in UTUC. Thus, we analyzed the methylation data by focusing on those involved in growth and identified INHBA as the most significant gene associated with cancer progression in UTUC.

According to previous studies, the upregulation of INHBA mRNA expression occurs after treatment with the methyltransferase inhibitor, 5-aza-2’-deoxycytidine, in esophageal adenocarcinoma and lung adenocarcinoma cell lines. INHBA overexpression may be affected by promoter methylation in cancer cells [23,24]. Aberrant methylation and differential expression of INHBA are associated with prognosis in gastric cancer, based on RNA-Seq and Illumina Human Methylation 27 Chip data from the TCGA database [26]. In this study, our prospectively collected UTUC and normal urothelial tissue samples further showed INHBA hypomethylation in UTUC using pyrosequencing. In addition, real-time PCR and immunohistochemical analysis revealed that the transcript and protein expression levels of INHBA were higher in UTUC than normal urothelium. INHBA hypomethylation and its overexpression in tumors are thus consistent with the findings of previous reports [23,24,25,26].

INHBA is reported to be involved in cell growth, proliferation, apoptosis, metastasis, and carcinogenesis [29,30,31,32,39]. INHBA is involved in various physiological processes through autocrine and paracrine functions [40,41]. In the current study, we showed that INHBA silencing reduced tumor growth and migration in BFTC-909 cells. Yu et al. reveal that INHBA promotes the epithelial–mesenchymal transition and accelerates the motility of breast cancer cells through the TGF-β signaling pathway [31]. INHBA enhances the proliferation, migration, and invasion of colon cancer cells by upregulating VCAN [30]. Collectively, previous and new results indicate that INHBA may have a tumor-promoting effect in a variety of cancers.

Hypomethylation and increased expression of INHBA in bladder cancer were identified in the TCGA dataset using the UALCAN web portal (Figure 2). INHBA expression was overexpressed in late tumor stages (T3–T4), high-grade, and non-papillary subtypes in bladder cancer. A similar phenomenon was also observed as INHBA expression increased in human urothelial cancer cell lines that belong to the high grade. The high-grade urothelial carcinoma cell lines (T24 and BFTC-909) expressed higher levels of INHBA than low-grade urothelial carcinoma cell lines (RT4 and UM-UC-14) by real-time PCR. We further investigated the differentially expressed genes and signaling pathways related to INHBA in the TCGA-BLCA cohort using the LinkedOmics database and the Enricher web tool. The expression of INHBA was positively correlated with SMAD binding and extracellular structure organization by GSEA. According to previous studies, INHBA regulates cell proliferation and migration by activating the SMAD signaling pathway [31,32,42]. Consistent with previous studies, our data indicate an association between INHBA expression and extracellular structure-related genes, such as FN1, VCAN, and COL1A1 [30,43,44].

Cancer-associated fibroblasts (CAFs) have been reported to influence tumor growth and migration by regulating ECM components in a variety of cancers [45,46]. INHBA production from tumor cells is necessary for CAF activation and increased collagen levels in CAFs [47]. The Smad2 signaling pathway is involved in INHBA-induced stromal fibroblast activation [48]. Kumar et al. uncovered distinct CAF subtypes with high INHBA-FAP cell populations as predictors of poor clinical prognosis [49]. Collectively, INHBA may play an important role in regulating the CAF phenotype in cancers.

Based on the present results, INHBA is hypomethylated and overexpressed in UTUC. INHBA depletion reduces tumor growth and migration in vitro. High INHBA expression was found to lead to relatively high expression levels of ECM marker genes such as FN1, COL1A1, COL3A1, VCAN, and COL5A1. High INHBA expression levels may facilitate activin A, the formation of INHBA homodimers, leading to activation of activin receptors in cancer [50]. The tumor-promoting effects of INHBA are similar in many other cancer studies. Further development of INHBA screening or therapeutic agents regulating INHBA methylation/expression may help with UTUC prevention or treatment.

## 4. Materials and Methods

### 4.1. Tissue Samples and Cell Line Study

RT4, T24, and BFTC-909 cell lines were purchased from the Bioresource Collection and Research Center (BCRC). The UM-UC-14 cell line was purchased from the European Collection of Authenticated Cell Cultures (ECACC). These cell lines were previously described in detail [20]. Tissue specimens of UTUC were collected from patients at the Chang Gung Memorial Hospital in Kaohsiung. The study was approved by the Institutional Review Committee of the Chang Gung Medical Foundation (IRB number: 201504731B0). Silencer™ Select Negative Control No. 1 siRNA (cat. no. 4390844), INHBA siRNA-A (cat. no. 4392420, siRNA ID: s7435), and INHBA siRNA-B (cat. no. 4392420, siRNA ID: s7436) were purchased from Thermo Fisher Scientific, Waltham, MA, USA. These siRNAs were dissolved at 10 μM in DNase and RNase-free water and stored in 10 μL aliquots at −80 °C until use. Lipofectamin RNAiMAX Transfection Reagent (cat. no. 13778150; Invitrogen, Thermo Fisher Scientific) was used to transfect 10 nM siRNA into BFTC-909 cells by incubation at 37 °C according to the manufacturer’s protocol, followed by incubation for 48 h prior to subsequent experimentation. We used a Boyden chamber for the migration assays. BFTC-909 si-control, si-INHBA (A), and si-INHBA (B) cells were seeded in the upper chamber. After 18 h of migration, cell migration was assessed. The cells on the lower side of the membrane were fixed and stained with crystal violet.

### 4.2. DNA Methylation Analysis

We used the QIAamp DNA Mini Kit (Qiagen, Hilden, Germany) to isolate DNA from clinical tissues. After passing the quality control criteria, the DNA samples were subjected to immunoprecipitation using proteins with a methyl-CpG-binding domain. Next, the enriched DNA fragments were amplified via PCR and loaded onto the GeneChip Human Promoter 1.0R tiling array (Affymetrix, Santa Clara, CA, USA). We evaluated the difference in methylation between tumor samples with high-grade and high-stage UTUC (*n* = 3) and normal urothelium adjacent to low-grade and low-stage UTUC (*n* = 3) by comparing the probe intensities of the promoter regions. A DNA methylation analysis was performed as described previously [19].

### 4.3. INHBA Promoter Methylation and Gene Expression Analysis of the Clinical Characteristic from the UALCAN Web

UALCAN is an online TCGA analysis database that links multi-omics data from TCGA datasets [33]. INHBA promoter methylation and mRNA expression in each BLCA patient were examined in TCGA datasets using the UALCAN web with default settings. Promoter methylation was analyzed according to sample type. Data from probes cg19535073, cg18413237, cg14527389, and cg16415646 in the Infinium Human Methylation 450 K chip were used for the promoter methylation data in the UALCAN web. INHBA mRNA expression in cancer was separately analyzed with regard to the patient characteristics of tumor stages, nodal metastasis status, histologic grade, histologic subtypes, and molecular subtypes. The plots and labels downloaded from UALCAN were modified for readability. The statistical analysis between two variables was performed by unpaired t-test. One-way ANOVA was performed for more than two variables.

### 4.4. Pyrosequencing-Based Bisulfite PCR Analysis

A total of 500 ng of DNA from each sample was treated using the EZ DNA Methylation-Lightning Kit Bisulfite Conversion System (Zymo Research, Irvine, CA, USA). The converted DNA was then eluted in 20 μL of elution buffer. Bisulfite conversion was performed in the dark at 98 °C for 10 min and 64 °C for 3.5 h, followed by desulfonation of the converted DNA. Gene amplification was performed using the HotStarTaq Master Mix Kit (Qiagen, Hilden, Germany). The INHBA pyrosequencing primers for CpG1 and CpG2 sites were: forward: 5’-ATTAAATATTTGGAGAGGAGGAAATTG-3’; reverse: 5’-AAAACAAACCAACTAAAACCTAAAC-3’; sequencing: 5’-AGGAAATTGATGGGATT-3’. The amplification conditions were as follows: 95 °C for 15 min, 94 °C for 30 s, 60 °C for 30 s, 72 °C for 30 s, and 45 cycles. To assay DNA methylation levels of the INHBA promoter, bisulfite sequencing was performed using the PyroMark Q24 instrument (Qiagen). The relative levels of methylation at each CpG site were analyzed using the PyroMark Q24 version 2.0.6.

### 4.5. RNA Isolation and Real-Time PCR

RNA was extracted from UTUC clinical tissues and cell lines using a QIAGEN RNA purification kit. One microgram of RNA from each sample was reverse transcribed using RevertAid^TM^ H Minus Reverse Transcriptase (Fermentas, Waltham, MA, USA). Real-time PCR was performed using SYBR Green PCR master mix (Life Technologies, Carlsbad, CA, USA) and an ABI 7500 sequence detection system (Life Technologies, Carlsbad, CA, USA). The real-time PCR primers were as follows: INHBA forward: 5’-GGATGACATTGGAAGGAGGGCA-3’; INHBA reverse: 5’-ACTGACAGGTCACTGCCTTCCT-3’. RPL37A forward: 5’-AATCAGCCAGCACGCCAAGTAC-3’; RPL37A reverse: 5’-GCCACTGTCTTCATGCAGGAAC-3’. FN1 forward: 5’-ACAACACCGAGGTGACTGAGAC-3’; FN1 reverse: 5’-GGACACAACGATGCTTCCTGAG-3’. COL1A1 forward: 5’-GATTCCCTGGACCTAAAGGTGC-3’; COL1A1 reverse: 5’-AGCCTCTCCATCTTTGCCAGCA-3’. All primers were purchased from OriGene (Rockville, MD, USA) and checked for specificity using BLAST (NCBI). Exon/intron junctions were spanned.

### 4.6. Immunohistochemistry

Immunostaining for INHBA was performed using a fully automated Bond-Max system (Leica Microsystems, Wetzlar, Germany). Slides carrying tissue sections cut from formalin-fixed, paraffin-embedded tissue microarray blocks were dried for 1 h at 60 °C. These slides were then covered by Bond Universal Covertiles and placed in the Bond-Max instrument. All subsequent steps were performed automatically by the instrument according to the manufacturer’s instructions (Leica Microsystems) as documented in a previous report [51]. Primary antibody reaction: INHBA (1:100; cat# sc-166503, Santa Cruz, CA, USA), and the reaction was conducted at RT for 60 min.

### 4.7. Western Blotting Assays

BFTC-909 cells were lysed in RIPA buffer containing a protease inhibitor mixture (Roche, Basel, Switzerland). For each lane of 10% SDS–PAGE gel, 30 μg of cell lysate protein was loaded, separated, and subsequently transferred onto Immobilon-P Transfer Membrane (Millipore, Burlington, MA, USA). The membranes were probed with specific antibodies including the primary antibodies against INHBA (cat# sc-166503; Santa Cruz, CA, USA; 1:500), SMAD4 (cat# sc-7966; Santa Cruz, CA, USA; 1:1000), SMAD3 (cat# ab40854; Abcam, Cambridge, UK; 1:1000), SMAD2 (cat# ab40855; Abcam, Cambridge, UK; 1:1000), and beta-actin (cat# ZRB1312; Sigma, St. Louis, MO, USA; 1:5000). The secondary antibodies were added and incubated for 1 h and visualized using chemiluminescence. Enhanced chemiluminescence Western blotting reagents were obtained from Pierce Biotechnology (Rockford, IL, USA).

### 4.8. Co-Expression Analysis of INHBA from the LinkedOmics Database and the Enricher Web Tool

The LinkedOmics database is mainly used for comprehensive data analysis from TCGA datasets across 32 types of cancer [34]. The LinkFinder module was used to analyze the differentially expressed genes related to INHBA in TCGA-BLCA cohort (*n* = 408). Co-expression of INHBA was analyzed statistically using the Pearson correlation coefficient and the results were visualized using a heat map. To identify gene ontology and transcription factors shared by INHBA-correlated genes from the LinkedOmics database, we used the Enricher web tool [35]. The enriched GO and transcription factor protein–protein interactions were visualized using a bar diagram. The LinkInterpreter module was used to perform GSEA to obtain the related GO analysis (biological processes). GSEA is a computational method that determines whether an a priori defined set of genes shows statistically significant, concordant differences between two biological states. The rank criterion was *p* < 0.05, and 500 simulations were tested.

### 4.9. Statistical Analysis

The statistical analysis for each experiment is described in the figure legends. All graphs and analyses were performed using GraphPad Prism 8.0.2 software (San Diego, CA, USA) and analyzed using one-way ANOVA followed by the Tukey’s multiple-comparison test or Student’s *t*-test. Results are expressed as the mean ± standard error of the mean (SEM).

## Figures and Tables

**Figure 1 ijms-23-02072-f001:**
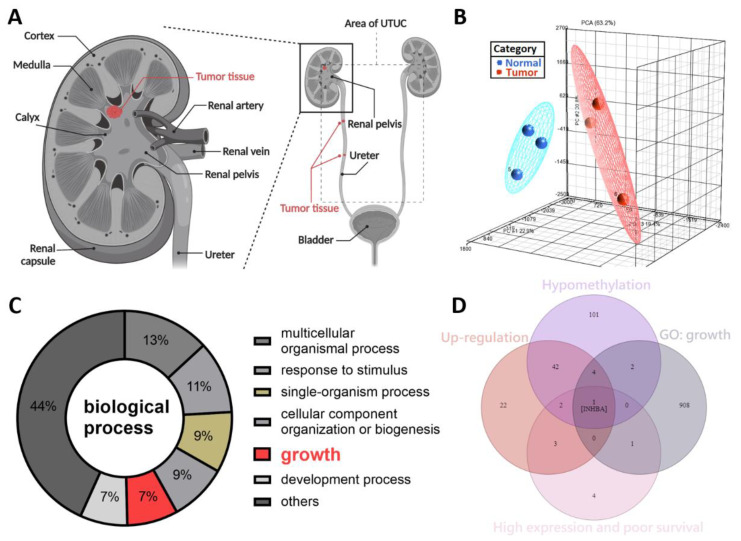
DNA methylation profiling analysis identified INHBA as a potential oncogene in UTUC. (**A**) Schematic illustration of UTUC referred to as renal pelvic and ureteral tumors. Created with BioRender.com (accessed on 29 December 2021). (**B**) Principal component analysis plot based on the methylation profiles for UTUC (red) and normal urothelium (blue). (**C**) Gene ontology analysis of hypo-DMR in biological process. Pie charts of gene categories (percentage indicated: multicellular organismal process: 13%; response to stimulus: 11%; single-organism process: 9%; cellular component organization or biogenesis: 9%; growth: 7%; development process: 7%). (**D**) Venn diagram depicting overlapping genes in four conditions. Condition 1 included 152 genes that have the hypomethylation of the promoters in tumor tissues. Condition 2 revealed 74 genes that have up-regulation in tumor tissues. Condition 3 contained 11 genes that have high expression and poor survival. Condition 4 had 916 genes associated with growth (GO: 40007).

**Figure 2 ijms-23-02072-f002:**
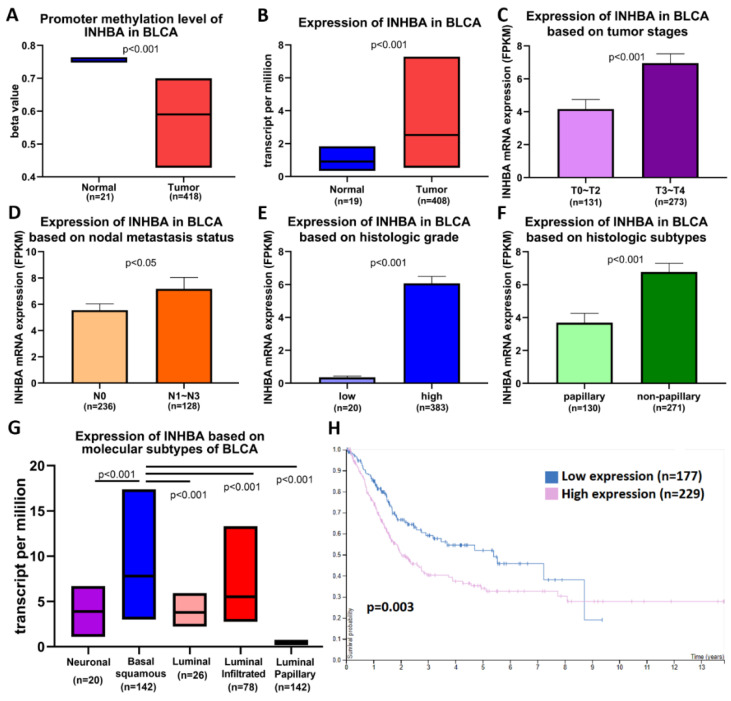
INHBA is hypomethylated and is a biomarker of tumor progression and metastasis in BLCA patients based on the UALCAN web tool. (**A**) Promoter methylation levels of the INHBA gene expressed from TCGA-BLCA clinical data. (**B**) Box plots showing the INHBA mRNA expression in BLCA tumors (red plot) and the normal (blue plot) tissues. The INHBA mRNA expression level was expressed for the patient characteristics of (**C**) tumor stages, (**D**) nodal metastasis status, (**E**) histologic grade, (**F**) histologic subtypes, and (**G**) molecular subtypes. (**H**) High expression of INHBA is associated with a lower survival rate in BLCA patients.

**Figure 3 ijms-23-02072-f003:**
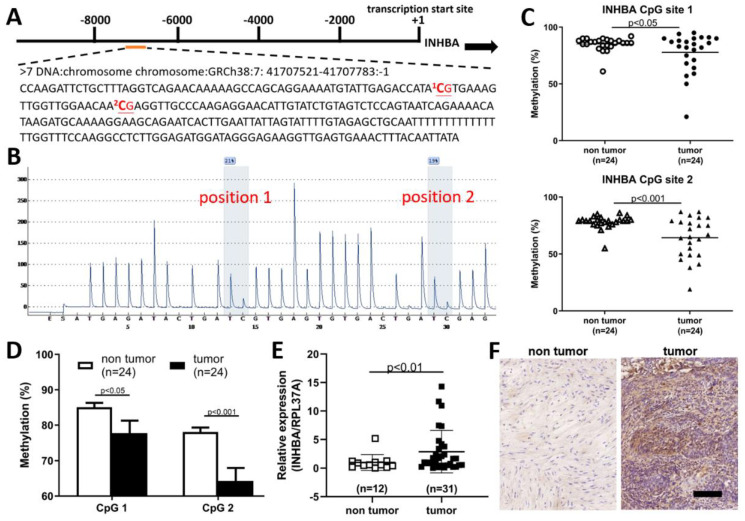
INHBA associated with hypo-DMR was up-regulated and correlated with tumor malignancy in upper urinary tract urothelial carcinoma. (**A**) Schematic of the INHBA locus. There were two CpG sites in the 5′ upstream region of INHBA; a total sequence of 263 base pairs (bp) is presented. (**B**) Two methylation sites in the 5′ upstream region of INHBA were identified by bisulfite pyrosequencing. (**C**,**D**) Low methylation in the CpG site of INHBA in tumor tissues compared to paired non-tumor tissues (adjacent urothelium, *n* = 24). (**E**) INHBA transcript levels as detected by real-time PCR were significantly upregulated in upper tract urothelial carcinoma. Data are denoted as mean ± SEM, the *p* values were calculated with Student’s *t*-test. (**F**) Immunohistochemical analysis revealed a significant increment in INHBA immunoexpression in tumor tissues. Scale bars indicate 100 μm.

**Figure 4 ijms-23-02072-f004:**
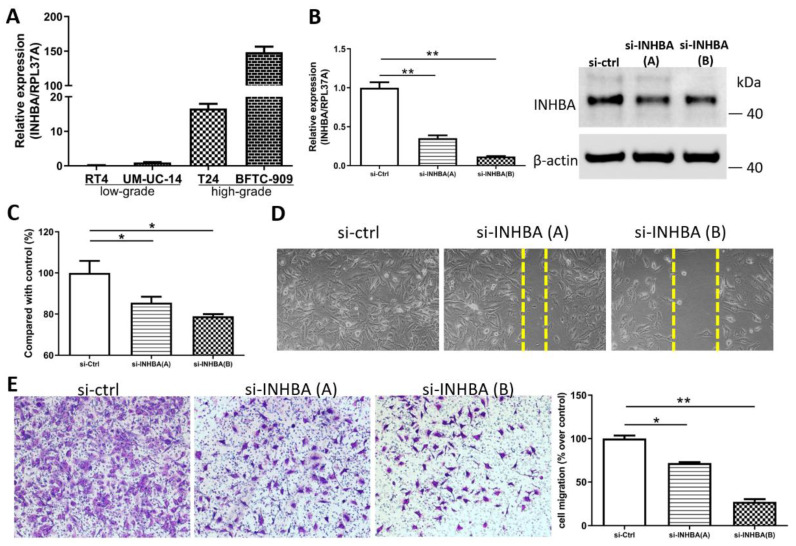
INHBA increases the proliferation and migration of BFTC-909 cells. (**A**) Real-time PCR of INHBA expression in 2 low-grade urothelial carcinoma cell lines (RT4 and UM-UC-14), and 2 high-grade urothelial carcinoma cell lines (T24 and BFTC-909). (**B**) BFTC-909 cells transfected with si-control and si-INHBA were detected by real-time PCR and Western blotting. (**C**) The viability of BFTC cells transfected with si-control and si-INHBA was determined using the alamarBlue assay. Error bars represent mean ± S.E.M., the *p* values were calculated with one-way ANOVA followed by Tukey’s multiple-comparison test, * *p* < 0.05 versus si-control group. (**D**) The cell migration of BFTC-909 cells transfected with si-control and si-INHBA was determined by the wound healing assay. (**E**) Inhibition of INHBA expression via siRNA repressed the migration ability of BFTC-909. Error bars represent mean ± S.E.M., the *p* values were calculated with one-way ANOVA followed by Tukey’s multiple-comparison test, * *p* < 0.05 versus si-control group, ** *p* < 0.01 versus si-control group.

**Figure 5 ijms-23-02072-f005:**
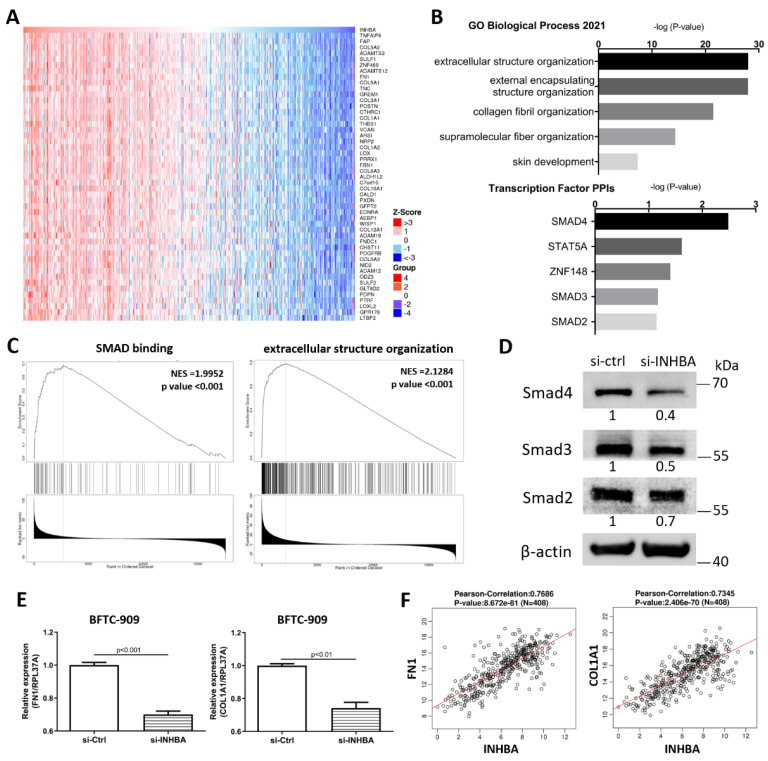
Co-altered genes profile with the INHBA gene involved in signaling pathways in urothelial carcinoma. (**A**) Heat map showing the top fifty genes positively correlated with INHBA in BLCA by LinkedOmics. (**B**) GO analysis (biological processes) and Transcription Factor protein–protein interactions of top fifty genes positively correlated with INHBA performed via Enricher. (**C**) GSEA showed that INHBA was closely associated with SMAD binding and extracellular structure organization. NES refers to normalized enrichment score. (**D**) The protein level of Smad4, Smad3, and Smad2 in BFTC-909 transfected with si-control and si-INHBA. actin was used as loading control in Western blot. (**E**) Real-time PCR of FN1 and COL1A1 expression in BFTC-909 transfected with si-INHBA. Error bars represent mean ± S.E.M., the *p* values were calculated with Student’s *t*-test. (**F**) FN1 and COL1A1 expression correlation with INHBA RNA expression in the TCGA BLCA database.

**Figure 6 ijms-23-02072-f006:**
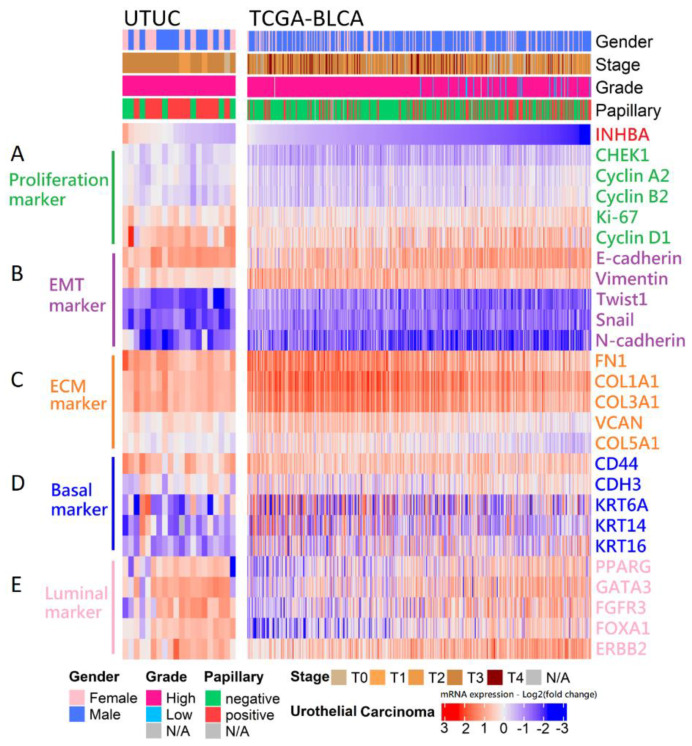
Analysis of the mRNA expression subtypes according to the level of INHBA in UC. Top, left to right: UTUC, 20 samples were analyzed using the unlinked data from the NHRI bioinformatics database for upper tract urothelial carcinoma. TCGA-BLCA, 408 samples were re-analyzed using the cBioPortal online tool for bladder urothelial carcinoma (TCGA, Firehose Legacy). Covariates: selected clinical covariates included gender, stage, grade, and papillary histology; log2 (fold change) for selected genes; gene sets labeled as (**A**) proliferation marker, (**B**) EMT marker, (**C**) ECM marker, (**D**) basal marker, and (**E**) luminal marker.

## Data Availability

Luo and Kao had full access to all data in the study and assume responsibility for the integrity of the data and the accuracy of the data analysis. The data presented in this study are available upon request from the corresponding author. The data are not publicly available owing to privacy restrictions.

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
