# Peer review of "DNA Hypomethylation Is Associated with the Overexpression of INHBA in Upper Tract Urothelial Carcinoma"

_ijms, 2022, doi:10.3390/ijms23042072_

Round 1

Reviewer 1 Report

The authors performed INHBA expression and DNA methylation analysis in urothelial carcinoma. By presenting an appropriate methods and clear results, the point was provided that hypomethylation and overexpression of INHBA were associated with a poor prognosis.

There are no problems with the overall story, analysis method, and conclusion. However, there are problems in some minor parts, and if it is corrected, it is expected that it can be published in IJMS.

  1. A brief schematic diagram of the anatomical location of the UTUC should be presented in figure 1. Although many studies have been conducted on urothelial carcinoma, it has received relatively little attention compared to major cancers such as lung, colon, and stomach cancer. Therefore, a schematic diagram of the anatomical location has been presented briefly, and it is necessary to mention which part was processed.
  2. Line 97: Duplicated "gastric cancer".
  3. It is unclear which three normal vs three cancer sample was used in Figure 1A. TCGA-BLCA?
  4. It is unclear why "Growth" is highlighted in Figure 1B.
  5. Line 153: real-time polymeragse chain reaction, full name required? It is enough to indicate by PCR, and the same should be displayed in the rest of the manuscript.
  6. Line 229: GESA -> GSEA. The description of GSEA is lacking in the manuscript. Why did you choose 2 terms from which set out of 7 gene sets? Statistical significance?
  7. Abbreviations: Overall, there are no rules. e.g. line 251 ECM, but line 321 extracellular matrix. Line 320 "Cancer-associated fibroblasts" -> "Cancer-associated fibroblasts (CAFs)"
  8. The heatmap in Figure 6 is an appropriate visualization strategy. Perfect in every thing, but what does the black border mean? Since there is no explanation at all, an appropriate explanation should be added or deleted.
  9. What is the analysis platform for visualization in Method? It should indicate which package or function was used on which platform. e.g. We used the pheatmap function of R's pheatmap package for heatmap and the XXXXX parameter.

Reviewer 2 Report

The authors using MBD protein microarray and pyrosequencing identify INHBA was induced in UTUC by DNA hypomethylation. They showed INHBA plays functions in proliferation and migration using gene knockdown in cell lines. Furthermore, using LinkedOmics database, the co-altered genes profile with INHBA was demonstrated, indicating INHBA was involved in the SMAD signaling pathway and extracellular structure organization. They provide evidence to prove DNA hypomethylation is associated with INHBA upregulation in high-grade UTUC. I have some suggestions.

  1. The authors mentioned that the symptoms of UTUC are non-specific and are challenging to detect at an early stage. Based on these descriptions, identifying a biomarker for early diagnosis is a more critical issue than identifying a therapeutic target. Do you try to analyze DNA methylation profiling using low-grade and low-stage UTUC samples?
  2. Although the experiment of methyl-CpG binding domain protein capture was performed in your previous paper, I suggest you include a brief description of this method in Materials and Methods.
  3. In Fig.3A, I suggest the authors use the transcription start site, not the translation initiation site (ATG), as +1 to label the sequence numbers. The -7354 region should not be referred to as “promoter”; it might be changed to “5’ regulatory region” or “5’ upstream region”.
  4. In Fig.4, I suggest the authors treat cells with DNA methylation inhibitors (such as decitabine) to monitor the INHBA expression. It will demonstrate INHBA is up-regulated by DNA hypomethylation.
  5. The lettering of Fig.5C is too small to recognize the results. I do not understand the description on lines 215-217. What are “SMAD binding” and “extracellular structure organization”? Could you explain more?

Round 2

Reviewer 2 Report

The manuscript has been improved. I agree to its publication on IJMS.